# Karst Dolines Support Highly Diversified Soil Collembola Communities—Possible Refugia in a Warming Climate?

**Michal Marcin** [1,*], **Natália Raschmanová** [1], **Dana Miklisová** [2], **Jozef Šupinský** [3], **Ján Kaňuk** [3]
**and Ľubomír Kováč** [1]

1   Institute of Biology and Ecology, Faculty of Science, Pavol Jozef Šafárik University in Košice,
    SK-040 01 Košice, Slovakia
2   Institute of Parasitology, Slovak Academy of Sciences, SK-040 01 Košice, Slovakia
3   Institute of Geography, Faculty of Science, Pavol Jozef Šafárik University in Košice, SK-040 01 Košice, Slovakia
*   Correspondence: michal.marcin@student.upjs.sk

**Abstract:** Karst dolines, as geomorphologically diverse natural landforms, usually exhibit more or less steep microclimatic gradients that provide a mosaic of diverse microhabitat conditions, resulting in a high diversity of soil biota with numerous rare endemic and/or relict species occupying these habitats. In this study, we investigated the spatial patterns of Collembola abundance, species richness, community structure and distribution of functional groups at topographically and microclimatically different sites across three open (unforested) karst dolines in a north–south direction in the Slovak Karst, Slovakia. We also assessed the refugial capacity of dolines for collembolan communities. The Friedman ANOVA test confirmed the significant differences in soil mean temperatures between the sites of all the dolines selected. The diverse soil microclimatic conditions within the dolines supported higher Collembola diversity (species numbers, diversity indices) compared with sites on the karst plateau and showed a potential to facilitate the persistence of some species that are absent or very rare in the surrounding landscape. In dolines with circular morphology and comparable size, the topography and soil microclimate had a stronger effect on community composition and structure than soil organic carbon. Shallow solution dolines provided microhabitats for various functional groups of soil Collembola in relation to the microclimatic character of the individual sites. It was observed that such landforms can also function as microclimatic refugia for cold-adapted species through the accumulation of colder air and the buffering of the local microclimate against the ambient mesoclimate, thus underlying the necessity of adequate attention in terms of the conservation of the karst natural phenomena.

**Keywords:** topography; microclimatic gradient; functional groups; $\alpha$-diversity; distributional pattern

## 1. Introduction

Climate warming is currently a major threat to the Earth's biodiversity [1,2]. Generally, climate change affects soil biota by altering environmental conditions through shifting the temperature distribution and mainly by changing the precipitation regime [3]. Recently, more attention has been paid to natural sites with specific microclimatic conditions that are considerably buffered from regional environmental changes and can provide a favourable and more stable microclimate during adverse climate periods [4]. These are geomorphologically diverse karstic and non-karstic landforms, such as gorges, cave entrances, ravines, valleys or surficial enclosed depressions ("dolines") that usually exhibit more or less steep microclimatic gradients. Several recent studies have focused on the effect of a meso-/microclimatic gradient on vegetation (e.g., [5–16]) and soil fauna (e.g., [8,17–28]) in such land formations. Previous studies have demonstrated that the specific microhabitat conditions of these landforms generally resulted in a high diversity of soil biota, including numerous stenothermal taxa, such as xerothermophilous/xeroresistant (adapted

to warm/dry conditions) and montane/psychrophilous species (adapted to cold/wet conditions), some of which were rare endemic and/or relict species.

Furthermore, such sites could potentially serve as microrefugia for terrestrial arthropods, where species adapted to cold and wet environmental conditions can survive for long time periods (e.g., [8,11,18,20,29–34]). There is a lack of long-term studies focusing on potential changes in soil arthropod communities in these delicate habitats in relation to climate warming, especially in the temperate zone (e.g., [28]). Given the fact that the soil fauna is highly resilient to climate change [28,35–39], such long-term studies are very important for predicting how a changing climate would affect functional biodiversity and how these reactions may affect ecosystem functioning [8,20,28,40,41].

In many karst areas, the dominant type of surficial enclosed depressions, i.e., solution dolines (also called sinkholes), are observed. Their diameters and depths range from metres to tens of metres, and their inner slopes vary from subhorizontal to nearly vertical [42]. The marked effect of microclimatic gradients on the structure of soil invertebrate communities and their functional (ecological) groups (according to temperature and moisture preferences), as well as the refugial capacity of such karst dolines, have already been documented in a limited sample of soil invertebrates belonging to macrofauna, including ants, beetles, snails, spiders and woodlice [8,21–23,26,27]. These studies have shown that karstic depressions, which have the potential to maintain permanently cold and wet conditions even during drier periods of the year, can provide shelter for rare and specific (cold-adapted) species, which especially dominate at the bottoms of the dolines. The soil microclimate and topography of dolines are important factors affecting the diversity and community structure of soil macrofauna in karst dolines and their surrounding plateau [27]. Moreover, dolines also host proportionally more small-bodied spiders and ground beetles with a higher dispersal capacity than a plateau, probably due to funnel-shaped topography capable of trapping good dispersers. These studies have thus shown that karst dolines can provide important habitats for many arthropods that are rare or absent in the surrounding plateau and can preserve distinctive arthropod communities, which contribute to the local biodiversity of karst environments on a small scale (α-diversity).

Despite the growing literature concerning "dolines", there is still a lack of research on soil mesofauna, such as mites, springtails and enchytraeids, which could shed more light on the distribution of the soil biota in these specific habitats. Collembola are among the most ecologically important members of the soil mesofauna and have been proposed as good model organisms for surveying the functional biodiversity of soils [43–45]. Many collembolan species are adapted to high and stable humidity and are sensitive to drought and temperature changes [46,47]. In terms of climate warming trends, Collembola are considered a reliable bioindicative group for local and regional climatic variations in soil environments (e.g., [48,49].

In the present study, we hypothesised that the collembolan community parameters would differ markedly at sites of open (unforested) karst dolines in association with their complex habitat conditions, such as topography, soil microclimate and soil-chemical parameters, and grassland vegetation associations. Regarding functional characteristics, we expected the distributional patterns of Collembola functional groups at selected sites across dolines to correspond with the local microclimate. We also aimed to assess the potential refugial capacity of karst dolines for Collembola. We expected the cold and wet bottoms of karst dolines with less pronounced soil microclimatic variations to serve as important microrefugia for relict taxa, surviving in these sites from cold glacial or postglacial periods. These taxa are considered highly vulnerable to climate warming in such fragile karst ecosystems. In this study, we aim at: (1) community analysis of soil Collembola at topographically and microclimatically different sites across three open karst dolines, (2) investigation of distributional patterns of functional groups of Collembola, and (3) evaluation of karst dolines in terms of their role as possible microrefugia for soil Collembola and their importance for the effective management and conservation of karst ecosystems under ongoing climate change.

## 2. Materials and Methods

### 2.1. Description of Study Area

The studied dolines were located in the south-western part of the Silická plateau in the Slovak Karst (Slovakia), which is a part of the Slovak-Aggtelek Karst geomorphological unit situated in Slovakia and Hungary (Figure 1A). The mean annual air temperature in the area ranges from 5.7 to 8.5 °C, and the mean annual precipitation ranges from 630 to 990 mm. This area has a typical karst character with various karst landforms, such as forested and unforested solution dolines, collapse dolines, pits and sinkholes, with a specific microclimatic regime characterised by a lesser or stronger temperature inversion. There are drier and warmer sites on the south-facing slopes of the dolines, which receive more insolation than other doline microhabitats during the daytime, in contrast to the cold and wet sites of the north-facing slopes and the lower parts of the dolines (bottoms), which are affected by the accumulation of cold air [8,50–52].

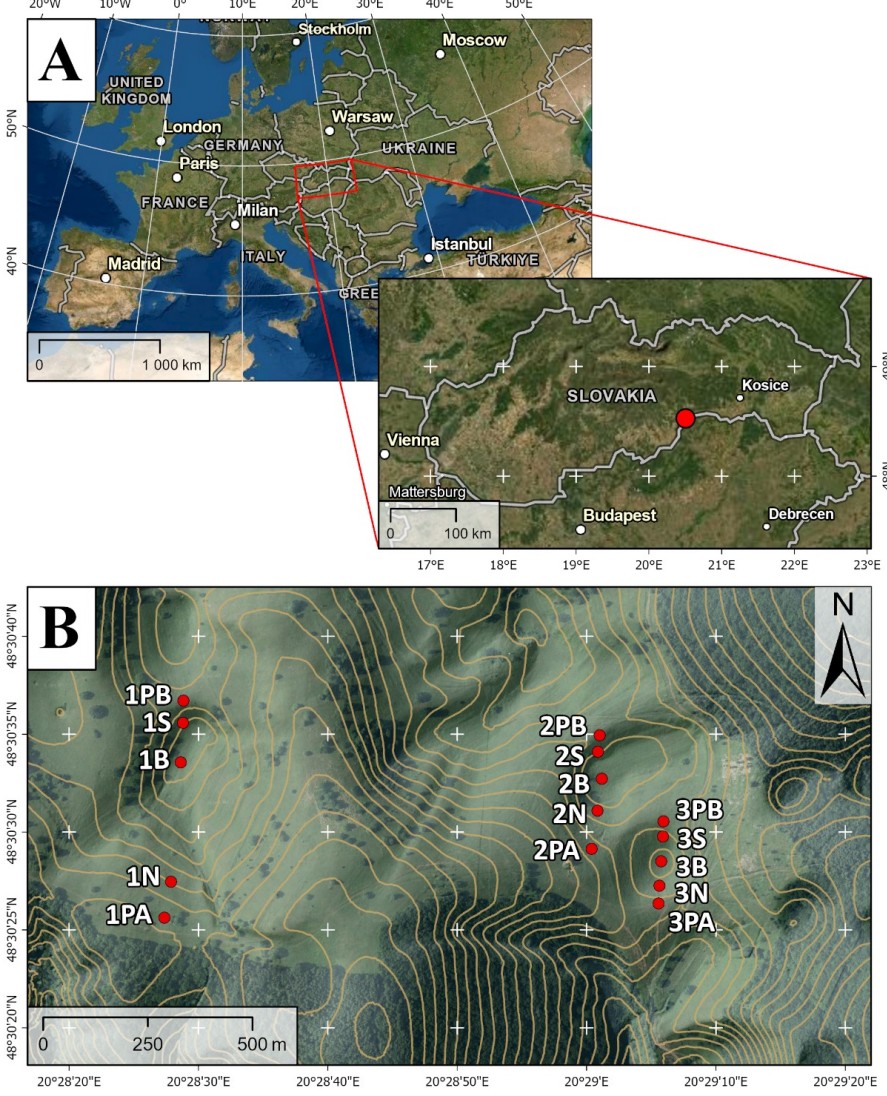

**Figure 1.** Study area, karst dolines and sites. (**A**) Location of Slovakia within Europe and location of the study area in Slovakia. (**B**) Detailed location of different sites along the doline (1–3); PA—plateaus south of the doline, N—N-facing slope, B—bottom of the doline, S—S-facing slope, PB—plateaus north of the doline. Contour step 5 m, Basemaps: Hybrid Imagery ©ESRI and Or-thophotomap ©ÚGKK SR.

### 2.2. Study Sites and Sampling Design

Three open dolines of different shapes, sizes and depths near the village of Kečovo in the Slovak Karst were selected for the study (Figures 1B and 2A–C). Doline (1) was slightly elliptical and larger compared to dolines (2) and (3), both with a similar circular shape. The longer diameter of doline (1) was 339 m, while the shorter was ~270 m, with a depth of 36 m. Dolines (2) and (3) had a diameter of 174.5 and 129.7 m and a depth of 17 and 7 m, respectively. The distance between dolines (1) and (2) was ~470 m, and ~15 m between dolines (2) and (3). All three dolines are situated in an area actively used for cattle grazing.

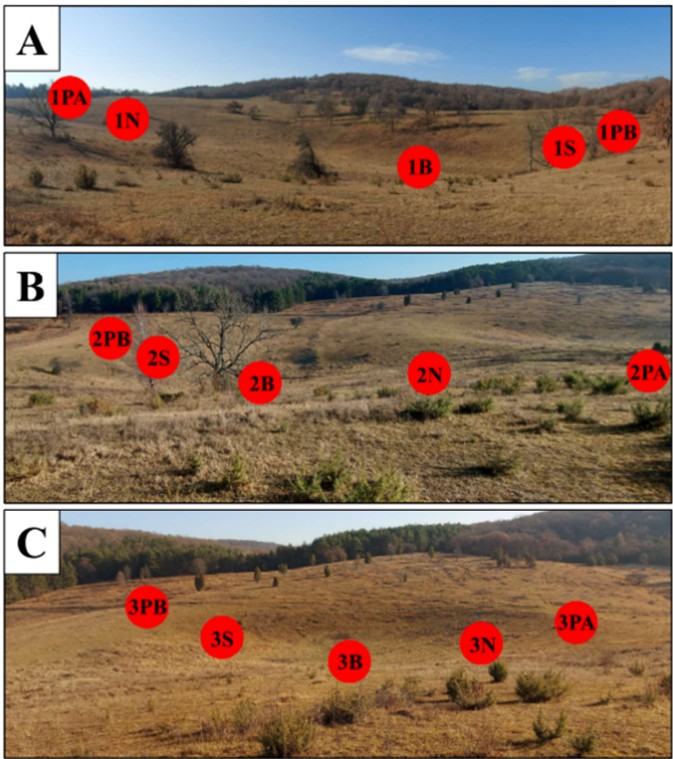

**Figure 2.** Detailed location of different sites along the doline (1–3); (**A**) Position of sites along the doline (1). (**B**) Position of sites along the doline (2). (**C**) Position of sites along the doline (3). (For site description, see Figure 1 and the Section 2).

Five sites were selected across each doline from south to north: PA—southern edge of the doline on the surrounding karst plateau, N—north-facing slope, B—doline bottom, S—south-facing slope, and PB—northern edge of the doline on the karst plateau. Transects across the dolines differed in length depending on the dimension of the individual doline (Figure 1B, Table 1). A total of five soil samples were taken from each site (75 in total from each doline) on 17 November 2019. The soil samples represented soil cores 10 cm in diameter, taken from a maximum depth of 8–10 cm (depending on the soil thickness). The samples were separated into plastic bags, transported to the laboratory, and extracted in a modified high-gradient apparatus [53] for 7 days. In order to prevent the death of fragile and sensitive species before the extraction, the temperature in the upper part of the box with samples was gradually increased from 15 to 55 °C during the procedure using an incandescent light as a heat source. Collembola individuals were sorted under a binocular Leica S6E stereomicroscope and identified under a Carl Zeiss Axiolab A1 phase-contrast microscope (Carl Zeiss Microscopy, Oberkochen, Germany) to the species level using multiple taxonomic keys (e.g., [54–63]). Collembola specimens are deposited in the collection of the Department of Zoology, Institute of Biology and Ecology, Pavol Jozef Šafárik University in Košice, Košice, Slovakia.

**Table 1.** Topographic, soil microclimatic and chemical characteristics of studied sites in three open dolines.

| Doline/Site | Coordinates | Distance [m] | Altitude [m a.s.l.] | Slope [°] | Exposition | Topographic Index | Solar Radiation [kWh.m$^{-2}$.year$^{-1}$] | Insolation [h.year$^{-1}$] | $T_{mean}$ [°C] | $T_{min}$ [°C] | $T_{max}$ [°C] | C [%] | N [%] | Soil pH$_{H2O}$ |
|---|---|---|---|---|---|---|---|---|---|---|---|---|---|---|
| 1PA | 48°30′24.71″ N, 20°28′21.87″ E | 0 | 468 | 3 | N | 7.08 | 952.93 | 2906.26 | 8.96 ± 7.00 | −0.25 | 23.25 | 4.98 | 0.46 | 6.09 |
| 1N | 48°30′26.55″ N, 20°28′22.39″ E | 56.5 | 460 | 12 | NE | 5.30 | 917.05 | 3616.94 | 9.75 ± 7.15 | 0 | 26.25 | 7.25 | 0.64 | 7.33 |
| 1B | 48°30′32.65″ N, 20°28′23.12″ E | 246 | 433 | 1 | N | 7.66 | 986.48 | 3225.26 | 9.11 ± 6.87 | −0.25 | 23.25 | 4.83 | 0.45 | 5.90 |
| 1S | 48°30′34.67″ N, 20°28′23.29″ E | 309 | 436 | 18 | SE | 4.52 | 1068.95 | 3098.40 | 10.08 ± 7.02 | −0.25 | 26.25 | 4.87 | 0.42 | 5.90 |
| 1PB | 48°30′35.79″ N, 20°28′23.33″ E | 339 | 442 | 2 | E | 3.60 | 1032.59 | 3635.64 | 9.57 ± 7.05 | −0.5 | 24.50 | 7.74 | 0.75 | 6.90 |
| 2PA | 48°30′28.23″ N, 20°28′54.88″ E | 0 | 432 | 7 | NE | 5.11 | 958.75 | 3664.84 | 9.27 ± 6.31 | 0 | 21.75 | 7.49 | 0.64 | 7.21 |
| 2N | 48°30′30.18″ N, 20°28′55.35″ E | 59.5 | 422 | 11 | NE | 6.59 | 911.19 | 3361.04 | 9.73 ± 6.78 | −0.5 | 24.50 | 8.16 | 0.72 | 6.91 |
| 2B | 48°30′31.81″ N, 20°28′55.69″ E | 104.5 | 416 | 1 | NE | 11.08 | 978.46 | 3102.45 | 9.00 ± 6.74 | −0.5 | 22.50 | 6.15 | 0.61 | 5.90 |
| 2S | 48°30′33.20″ N, 20°28′55.36″ E | 148.5 | 421 | 18 | S | 3.91 | 1143.93 | 3538.48 | 11.63 ± 7.09 | 1 | 26.75 | 9.34 | 0.91 | 7.24 |
| 2PB | 48°30′34.03″ N, 20°28′55.50″ E | 174.5 | 424 | 4 | SE | 4.20 | 1058.86 | 3757.31 | 10.42 ± 6.83 | 0 | 26.25 | 8.81 | 0.87 | 6.80 |
| 3PA | 48°30′25.44″ N, 20°29′00.06″ E | 0 | 426 | 3 | N | 5.04 | 1000.38 | 3670.67 | 10.05 ± 7.00 | 0 | 26.00 | 10.50 | 1.07 | 7.40 |
| 3N | 48°30′26.35″ N, 20°29′00.12″ E | 30 | 422 | 9 | N | 5.73 | 932.86 | 3596.80 | 9.56 ± 7.31 | −0.25 | 25.75 | 7.78 | 0.67 | 7.41 |
| 3B | 48°30′27.59″ N, 20°29′00.26″ E | 66.7 | 419 | 0 | N | 13.32 | 1032.42 | 3578.97 | 9.57 ± 6.61 | 0 | 22.50 | 8.00 | 0.72 | 6.10 |
| 3S | 48°30′28.87″ N, 20°29′00.41″ E | 103.7 | 423 | 10 | S | 5.90 | 1113.83 | 3654.93 | 10.90 ± 7.36 | 0 | 26.75 | 7.68 | 0.66 | 6.81 |
| 3PB | 48°30′29.63″ N, 20°29′00.45″ E | 129.7 | 427 | 5 | SW | 4.84 | 1070.80 | 3736.66 | 10.32 ± 6.65 | 0.5 | 26.00 | 10.60 | 0.93 | 7.04 |

$T_{mean}$—mean soil temperature and standard deviation, $T_{min}$—daily minimum soil temperature, $T_{max}$—daily maximum soil temperature, C—organic carbon, N—total nitrogen (For site description, see Figure 1 and the Section 2).

### 2.3. Soil Topographic, Vegetation, Microclimatic and Chemical Data

The coordinates of the locations in the dolines within the study sites were recorded in the national coordinate system (S-JTSK) (EPSG: 5514) by the global navigation satellite system (GNSS) using a Topcon HYPER HR receiver with a connection to the reference network of the Slovak real-time positioning service (SKPOS) in real-time kinematic (RTK) mode. To provide information on the topography of the individual sites in the dolines, data on the elevation, slope, exposition and topographic index, solar radiation and insolation were recorded. The topographic index is directly related to the probability of water accumulation at a given site [64], and it is used as a proxy for soil moisture estimates in the present study. In this study, the parameter of solar radiation is defined as the total annual solar radiation received at the soil surface, and insolation represents the annual number of hours of direct solar radiation at a given site in clear-sky conditions. Geomorphometric parameters were derived from the digital terrain model (DTM) using ArcGIS Pro (Spatial Analyst toolbox) and GRASS GIS (r.slope aspect module) software. As the computational inputs, 1-metre resolution (DTM) and the digital surface model (DSM—vegetation cover included) were used. Digital models were derived by the linear interpolation of the classified point cloud from the airborne laser scanning (provided by the Geodesy, Cartography and Cadastre Authority of the Slovak Republic) with a declared horizontal accuracy of 0.08 m and a vertical accuracy of 0.09 m. Solar radiation was modelled using the Area Solar Radiation tool in ArcGIS Pro software, with outputs of total radiation and insolation in clear-sky conditions for the whole year for the DSM inputs.

Vegetation associations at the sites were characterised according to [65]:

Doline (1), both plateau sites associated with the *Ranunculo bulbosi-Arrhenatheretum elatioris* Ellmauer in Mucina et al., 1993 association, relatively dense growth of shrubs at site (1PA), both slopes associated with the *Brachypodio pinnati-Molinietum arundinaceae* Klika 1939 association, and the bottom with typical vegetation of doline bottoms in karst areas with low elevation, the *Festuco rupicolae-Nardetum strictae* Dostál 1933 association;

Doline (2), the plateau sites associated with the *Onobrychido viciifoliae-Brometum erecti* (Scherrer 1925) Müller 1966 association, the N-facing slope associated with the *Scabioso ochroleucae-Brachypodietum pinnati* Klika 1933 association, the S-facing slope with thermophilous vegetation, the *Rosetum gallicae* Kaiser 1926 association, and the bottom contained relatively high and dense growths of grasses with characteristic cold vegetation of the *Alchemillo-Arrhenatheretum elatioris* Sougnez and Limbourg 1963 association;

Doline (3), the plateau sites associated with the *Festuco rupicolae-Caricetum humilis* Klika 1939 association, the N-facing slope associated with the *Onobrychido viciifoliae-Brometum erecti* (Scherrer 1925) Müller 1966 association, the S-facing slope with the *Scabioso ochroleucae-Brachypodietum pinnati* Klika 1933 association, and the bottom with the *Pastinaco sativae-Arrhenatheretum elatioris* Passarge 1964 association.

Soil temperature was measured continually at the sites every 4 h by data loggers exposed to 3 cm of soil depth from 8 November 2020 to 7 November 2021. For each site, the annual mean ($T_{mean}$), minimum ($T_{min}$) and maximum ($T_{max}$) soil temperatures were calculated. The differences in soil temperature ($T_{mean}$) between sites for each doline were tested using a Friedman ANOVA test, which confirmed significant differences in soil mean temperatures between the sites for dolines (1), (2) and (3) ($\chi^2 = 720.7$, N = 365, df = 4, $p < 0.00001$), ($\chi^2 = 1126.4$, N = 365, df = 4, $p < 0.00001$), ($\chi^2 = 609.3$, N = 365, df = 4, $p < 0.00001$), respectively.

The soil-chemical parameters were analysed at each doline site (Soil Science and Conservation Research Institute, Bratislava, Slovakia). Soil $pH_{H2O}$ was measured potentiometrically using a glass electrode and a reference calomel electrode as an active pH in water [66]. Organic carbon ($C_{OX}$) and total nitrogen ($N_{TOT}$) content [67] were measured on a CHNS-O Elemental Euro EA 3000 analyser (Italy) according to [68].

*2.4. Community Data*

Several principal parameters were calculated in order to characterise the Collembola communities at the sites of the studied dolines: mean abundance and species richness as quantitative parameters, and Shannon diversity and Pielou equitability indices as qualitative parameters. The differences in abundance means and species richness between individual sites were tested using the Kruskal–Wallis and post-hoc test [69] for each doline separately. The relationships between environmental factors and community parameters were estimated for each doline separately using the nonparametric Spearman's correlation coefficient [69]. Species with dominance (D) $\geq$ 3% were considered numerically dominant [70].

Similarities in the communities between the dolines and sites were analysed using non-metric multidimensional scaling (NMS) ordination based on species abundance at the sites. Autopilot with slow and thorough mode and Sörensen (Bray–Curtis) distance (recommended for community data) were selected. After randomisation runs, a 3-dimensional solution was accepted as optimal. NMS analysis was performed by the PC-ORD 7 package [71,72]. Species present in fewer than two individuals in the whole material were excluded from this analysis due to their low explanatory value.

Based on the experience of the authors and literature data, Collembola species were differentiated into six functional groups according to their habitat preferences in relation to: (1) soil moisture, i.e., hygrophilous (prefer moist/wet conditions), mesophilous (intermediate moisture conditions), xerophilous/xeroresistant (adapted to drier conditions), and (2) temperature, i.e., eurythermic (large temperature valency), cold-adapted (prefer cold conditions) and thermophilous species (warm conditions) (see Appendix A). The distributional patterns of the functional groups (species numbers, abundance means) were tested using General Linear Model (GLM) analysis, and four graphs were used to illustrate the distribution of the functional groups at the different sites; specifically, three models for hygrophilous, mesophilous and xerophilous species were built for moisture and one, related to thermophilous species, for temperature. The models were not analysed for the following functional groups: (1) species with a wide temperature tolerance (eurythermic) due to their low informative value in terms of the research aims, and (2) cold-adapted species, since only one species was assigned to this group. In these models, the abundance of the functional groups was treated as a dependent variable, the site as a fixed factor, and the location (dolines 1, 2, 3) as a random factor. For every functional group, the significance of the differences between the sites was tested using the Fisher LSD test [69].

## 3. Results

The observed values for the topographic, soil temperature and edaphic parameters at the sites are summarised in Table 1. More variable temperature values were observed at warm sites (1N, 2S, 3S), whereas the cold sites of the dolines (1B, 2PA, 3B) showed rather slight temperature variations (Figure 3). The Friedman ANOVA test confirmed the significant differences in the soil mean temperatures between the sites of dolines (1), (2) and (3) ($\chi^2 = 720.7$, N = 365, df = 4, $p < 0.00001$), ($\chi^2 = 1126.4$, N = 365, df = 4, $p < 0.00001$), ($\chi^2 = 609.3$, N = 365, df = 4, $p < 0.00001$), respectively.

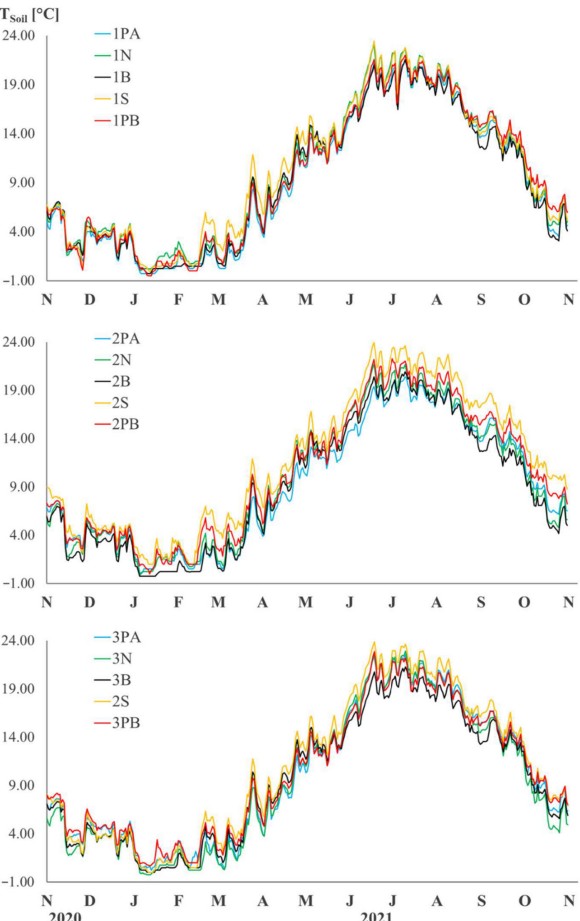

**Figure 3.** Temperature regime at different sites along karst dolines 1–3 (daily averages, $T_{Soil}$- soil temperature). (For site description, see Figure 1 and the Section 2).

A total of 8070 individuals and 63 species of Collembola were recorded at the studied sites (Appendix A). The abundance means of the communities at the sites across the dolines varied considerably, 6828–11,822 ind.m$^{-2}$ in doline (1), 10,548–20,153 ind.m$^{-2}$ in (2) and 3032–30,905 ind.m$^{-2}$ in (3). The number of species in the dolines ranged from 20 to 27 (1), from 13 to 28 (2) and from 20 to 29 (3) (Table 2).

**Table 2.** Community parameters of soil Collembola at studied sites in three open dolines.

| Doline/Site | A | S | H | J |
|---|---|---|---|---|
| 1PA | 11,822 ± 6878 | 27 | 2.70 | 0.82 |
| 1N | 7924 ± 1440 | 20 | 2.03 | 0.68 |
| 1B | 9427 ± 7980 | 21 | 2.37 | 0.78 |
| 1S | 9121 ± 4035 | 22 | 2.38 | 0.77 |
| 1PB | 6828 ± 1662 | 26 | 2.56 | 0.79 |
| 2PA | 20,153 ± 14,810 | 28 | 2.54 | 0.76 |
| 2N | 18,268 ± 4224 | 25 | 2.00 | 0.62 |
| 2B | 11,694 ± 2131 | 27 | 2.63 | 0.80 |
| 2S | 10,548 ± 2541 | 23 | 1.93 | 0.61 |
| 2PB | 12,102 ± 2078 | 13 | 0.97 | 0.38 |
| 3PA | 3032 ± 1453 | 20 | 2.19 | 0.73 |
| 3N | 11,032 ± 5876 | 22 | 2.36 | 0.76 |
| 3B | 30,905 ± 19,300 | 27 | 1.68 | 0.51 |
| 3S | 26,395 ± 9695 | 29 | 1.67 | 0.50 |
| 3PB | 16,357 ± 7546 | 20 | 1.42 | 0.48 |

A—abundance mean (ind.m$^{-2}$) ± standard deviation, S—species richness, H—Shannon diversity index, J—Pielou equitability index of evenness. (For site description, see Figure 1 and the Section 2).

Eleven species occurred exclusively at sites inside the dolines (N, B, S): *Entomobrya quinque-lineata*, *Friesea mirabilis*, *Friesea truncata*, *Mesaphorura rudolfi*, *Orchesella bifasciata*,

*Orchesella multifasciata*, *Proisotoma brevidens*, *Protaphorura serbica*, *Pseudosinella* cf. *csafordi*, *Willowsia buski* and *Willowsia nigromaculata*. On the other hand, 10 species were recorded exclusively at the plateau sites (PA, PB): *Axenyllodes bayeri*, *Doutnacia mols*, *Entomobrya handschini*, *Hemisotoma thermophila*, *Isotomodes productus*, *Karlstejnia annae*, *Micranurida pygmaea*, *Orthonychiurus rectopapillatus*, *Pumilinura loksai* and *Sminthurus maculatus* (Appendix A). The species *F. truncata* and *O. multifasciata* were the most abundant in the dolines and especially preferred their bottoms (B), while thermophilous and xerothermophilous *H. thermophila* and *I. productus* were characteristic for the plateau sites.

Two Carpathian/Western Carpathian endemics were documented at plateau sites 3PA and 1PA: *O. rectopapillatus* and the relatively abundant *P. loksai*. The first represents the only cold-adapted species found in the present study (Appendix A).

In doline (1), the highest values of the community parameters were recorded at the colder plateau site 1PA, while these values were lowest at the N-facing site 1N, except for the abundance means, which were lowest at the plateau site 1PB (Table 2). Considerably high mean abundances of three species, specifically *Isotomiella minor*, *Megalothorax minimus* and *Pseudosinella albida*, were recorded at site 1PA (Appendix A), as a result of aggregation in the occurrence of *I. minor* and *M. minimus* adults and *P. albida* juveniles, with high numbers in a few soil samples. In doline (2) the highest mean abundance and species richness were recorded at plateau site 2PA, while these values were the lowest at the warmer sites 2S and 2PB, respectively. The diversity indices were clearly the highest at the cold bottom site 2B. Abundant *Doutnacia xerophila*, *Mesaphorura critica* and *Protaphorura pannonica* were highly associated with plateau site 2PA. Aggregated distributions were observed in adults of *M. critica* and juveniles of *D. xerophila* and *P. pannonica*. The lowest diversity indices were observed at the warmer plateau site 2PB. Regarding doline (3), the highest values of mean abundance and diversity indices were recorded at the cold sites (3B, 3N), while species richness was the highest at the warm site 3S, with *I. minor*, *M. minimus* and *P. pannonica* as the most abundant species. The lowest mean abundance and species richness were noted at the plateau site 3PA and the lowest indices at the warm plateau site 3PB. Significant differences in the abundance means were confirmed by Kruskal–Wallis ANOVA only for doline (3) [H (4, N = 25) = 15.9, $p = 0.003$] between sites 3PA, 3B and 3PA, 3S. Significant differences in species richness were confirmed for dolines (2) and (3), i.e., between sites 2PA, 2PB; 2B, 2PB [H (4, N = 25) = 13.8, $p = 0.008$] and 3PA, 3S [H (4, N = 25) = 15.0, $p = 0.005$].

Several significant correlations were revealed between environmental factors and community parameters. In doline (1), the abundance of *Hypogastrura assimilis* positively correlated with $N_{TOT}$ (R = 0.90, $p = 0.04$) and $pH_{H2O}$ (R = 0.97, $p \leq 0.01$) and *P. pannonica* with $C_{OX}$ (R = 0.90, $p = 0.04$). In doline (2), the species richness negatively correlated with $T_{max}$ (R = −0.90, $p = 0.04$) and diversity indices with $C_{OX}$ and $N_{TOT}$ (R = −0.90, $p = 0.04$). The abundance of the dominant species *I. minor* negatively correlated with $T_{min}$ (R = −0.88, $p = 0.05$), the abundance of *Lepidocyrtus cyaneus* with $T_{min}$ (R = −0.95, $p = 0.01$) and *Parisotoma notabilis* with $C_{OX}$, $N_{TOT}$ and $T_{mean}$ (R = −0.9, $p = 0.04$). In doline (3), the mean abundance of Collembola negatively correlated with the soil $pH_{H2O}$ (R = −0.90, $p = 0.04$), and the diversity indices with $T_{min}$ (R = −0.89, $p = 0.04$). The mean abundance of *P. notabilis* negatively correlated with $T_{mean}$ (R = −0.90, $p = 0.04$) and $T_{min}$ (R = −0.89, $p = 0.04$) and *L. cyaneus* with $N_{TOT}$ (R = −0.90, $p = 0.04$).

Collembola communities at doline sites (1–3) were analysed using NMS ordination. The best three-dimensional solutions had a final stress of 7.34, $p < 0.00001$ after 49 iterations, which was confirmed by a Monte Carlo permutation test with $p = 0.03$ and a mean stress of 6.44 for real data and 250 runs for both real and randomised data. The first and second axes explained 40.1% and 32.4% of the variance. The NMS diagram suggested two clearly delimited clusters with respect to the microclimatic character of sites (Figure 4). The first cluster included species associated with the warmer sites of dolines (2) and (3) (2S, 2PB, 3S, 3PB), with characteristic (abundant) species *H. assimilis*, *Proisotomodes bipunctatus* and *Schoettella ununguiculata*. The second cluster represented other sites of the dolines, also

including the colder sites on the left side of the diagram (1B, 1PA, 2PA, 2B, 3N), associated with characteristic species *L. cyaneus*, *M. minimus*, *P. notabilis*, *P. pannonica* and *P. albida*. Sites 3B and 3PA were separated from both mentioned clusters. Moreover, it is shown in the ordination diagram that the Collembola communities of neighbouring dolines (2) and (3) had more similar community composition compared to doline (1).

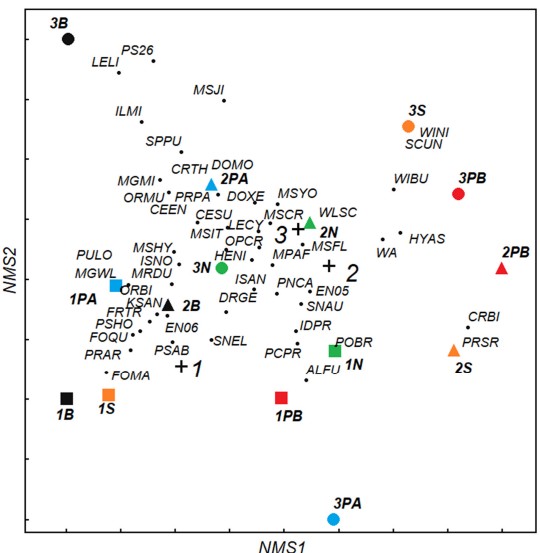

**Figure 4.** Non-metric multidimensional scaling (NMS) ordination for collembolan communities at different sites along dolines 1–3; blue—plateaus south of the dolines, green—N-facing slopes, black—bottoms of the dolines, orange—S-facing slopes, red—plateaus north of the dolines (For site description, see Figure 1 and the Section 2); Code of species: ALFU—*Allacma fusca*, CEEN—*Ceratophysella engadinensis*, CESU—*Ceratophysella succinea*, DRGE—*Desoria germanica*, DOMO—*Doutnacia mols*, DOXE—*Doutnacia xerophila*, EN05—*Entomobrya* sp. 1, EN06—*Entomobrya* sp. 2, FOMA—*Folsomia manolachei*, FOQU—*Folsomia quadrioculata*, FRTR—*Friesea truncata*, CRTH—*Hemisotoma thermophila*, HENI—*Heteromurus nitidus*, HYAS—*Hypogastrura assimilis*, ISAN—*Isotoma anglicana*, ILMI—*Isotomiella minor*, IDPR—*Isotomodes productus*, KSAN—*Karlstejnia annae*, LECY—*Lepidocyrtus cyaneus*, LELI—*Lepidocyrtus lignorum*, MGMI—*Megalothorax minimus*, MGWL—*Megalothorax willemi*, MSCR—*Mesaphorura critica*, MSFL—*Mesaphorura florae*, MSHY—*Mesaphorura hylophila*, MSIT—*Mesaphorura italica*, MSJI—*Mesaphorura jirii*, MSYO—*Mesaphorura yosii*, MPAF—*Metaphorura affinis*, MRDU—*Microgastrura duodecimoculata*, OPCR—*Oncopodura crassicornis*, ORBI—*Orchesella bifasciata*, ORMU—*Orchesella multifasciata*, ISNO—*Parisotoma notabilis*, PNCA—*Pratanurida cassagnaui*, POBR—*Proisotoma brevidens*, CRBI—*Proisotomodes bipunctatus*, PRAR—*Protaphorura armata*, PRPA—*Protaphorura pannonica*, PRSR—*Protaphorura serbica*, PCPR—*Pseudachorutes pratensis*, PSAB—*Pseudosinella albida*, PS26—*Pseudosinella* cf. *csafordi*, PSHO—*Pseudosinella horaki*, PULO—*Pumilinura loksai*, SCUN—*Schoettella unuguiculata*, SNAU—*Sminthurinus aureus*, SNEL—*Sminthurinus elegans*, SPPU—*Sphaeridia pumilis*, WA—*Wankeliella* sp. juv., WLSC—*Willemia scandinavica*, WIBU—*Willowsia buski*, WINI—*Willowsia nigromaculata*.

Graphs for the number of species occurrences and mean abundance showed several characteristic distributional patterns of functional groups of Collembola at topographically and microclimatically different doline sites (1–3) (Table 3, Figure 5, Appendix A). Comparing all sites together, significant differences in the species numbers were not confirmed only in the thermophilous group. However, regarding mean abundance, all the comparisons were non-significant (Table 3). Comparing pairs of sites, functional groups of Collembola showed significant preferences for certain sites. Regarding both species numbers and mean abundance, the hygrophilous group showed a preference for sites with colder conditions, specifically plateau sites (PA) and bottoms (B) that were significantly different from slopes and plateau sites with warmer conditions (S, PB). The mesophilous group showed a sig-

nificant preference for the colder bottoms of the dolines (B), significantly different from the other sites in species numbers and significantly different from the plateau sites (PA, PB) in mean abundance. The xerophilous/xeroresistant group showed a preference for plateau sites and warm, S-facing slopes (PA, S), with both parameters significantly different from the colder doline bottoms (B). Regarding species numbers, the thermophilous group showed a preference for warm S-facing slopes (S) and plateau sites (PA) compared to colder bottoms (B) and also showed preferences, although not significant, for warmer plateau sites (PB). Regarding mean abundance, this group clearly preferred the plateau sites (PA), but the differences with other sites were not significant.

**Table 3.** Comparisons of the number of species and mean abundances of Collembola functional groups for moisture (hygrophilous, mesophilous, xerophilous/xeroresistant) and temperature (thermophilous) requirements at sites across three open dolines (S—facing slopes, bottoms, N—facing slopes of dolines and plateau sites), GLM models.

|  | Hygrophilous | Mesophilous | Xerophilous/Xeroresistant | Thermophilous |
|---|---|---|---|---|
|  | F; *p* | F; *p* | F; *p* | F; *p* |
| Number of species | 3.26; **0.017** | 7.15; **<0.001** | 4.01; **0.005** | 1.79; 0.141 |
| Mean abundance | 2.38; 0.060 | 1.81; 0.136 | 2.07; 0.094 | 1.20; 0.319 |

(F(4, 68); *p*-value) Significant differences indicated by *p*-values in bold.

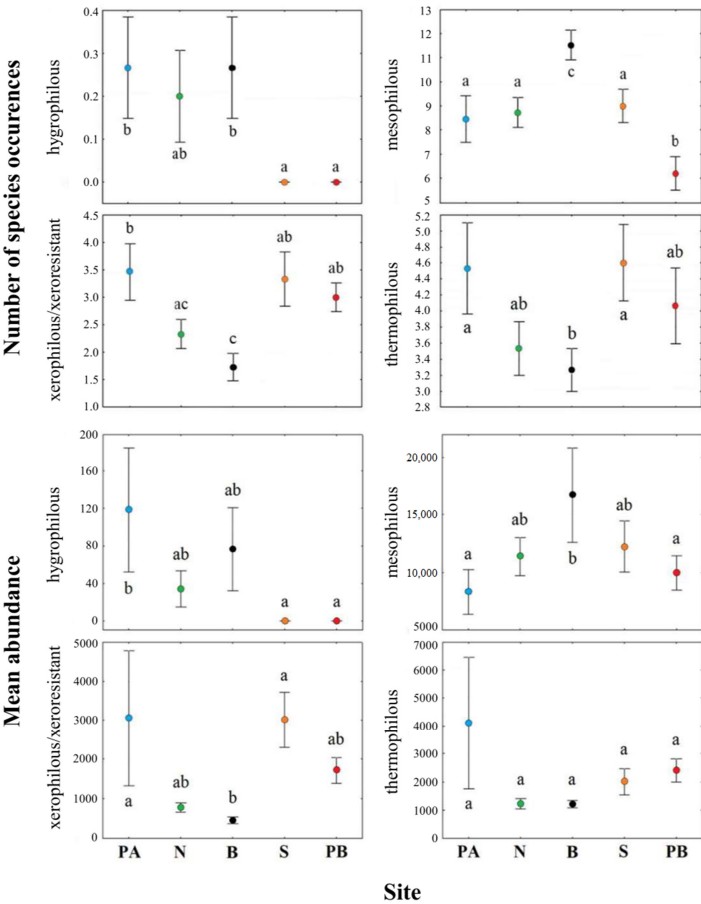

**Figure 5.** Number of species occurrences and abundance of Collembola (mean ± SE) belonging to different functional groups of moisture (hygrophilous, mesophilous, xerophilous/xeroresistant) and temperature (thermophilous) requirements at different sites (PA—plateaus south of the doline, N—N-facing slopes, B—bottoms of the doline, S—S-facing slopes and PB—plateaus north of the doline). Significant differences are indicated by different lowercase letters.

## 4. Discussion

### 4.1. Characteristics of the Sites at the Dolines

In the present study, the bottoms of dolines located at low elevations with a minimum slope inclination were considered the cold and wet habitats, which were observed in specific topographic and temperature characteristics. Doline bottoms received low insolation with a high topographic index (as a proxy for soil moisture estimate in the present study), indicating a higher probability of water accumulation, and had the lowest mean soil temperatures. In all the studied dolines, the highest value of the solar radiation was recorded on the south-facing slopes (S) and the lowest on the north-facing slopes (N). Regarding the larger and elliptically shaped doline (1), plateau site 1PA, with N/NE exposition and associated with several/sparse shrub growths near the adjacent forest, it showed a slightly lower temperature mean compared to its bottom and also received a markedly lower amount of insolation. It is known that local vegetation alters the temperature regimes at fine scales [73,74], and the presence of trees and shrubs may strongly reduce differences in insolation between topographically different sites [75]. Generally, N-facing slopes showed the lowest values of solar radiation and were slightly warmer than their cold doline bottoms, and in the case of dolines (1) and (2), they were also warmer than their plateau sites (PA). Sites on steep, S-facing slopes were associated with relatively low topographic indices, high solar radiation, high soil temperature means and a relatively high temperature amplitudes ($T_{max}$).

As expected, we observed that the doline bottoms and their N-facing slopes were in the principal pattern colder than the S-facing slopes, which is in congruence with other studies on this topic [8,23,74,76]. In addition to site topography and microclimate, doline bottoms contained a low amount of soil organic carbon, while the highest content of organic carbon was recorded at relatively warm plateau sites of the dolines (PB). The low soil carbon content in the cold and wet doline bottoms may be attributed to nutrient leaching, which increases with humidity [77].

### 4.2. Effect of Complex Habitat Conditions on Communities

Microclimatic and habitat heterogeneity are considered to be important factors influencing the structure and biodiversity of local plant communities [78,79] and animals [8,27,28,80,81]. Recent studies have shown that the topography and microclimate of solution karst dolines may considerably affect the communities of animals in terms of their abundance, species richness and community structure. It was observed that the abundance and diversity of woodlice and snails increased towards the lower parts of the dolines due to the more favourable moisture conditions [21–23]. Bátori et al., 2022 [27] documented that the surrounding plateau had higher species richness and abundance of ants and spiders compared to the doline, whereas woodlice and beetles showed higher values of these parameters in the doline. Indeed, the diversity and abundance patterns across karst dolines cannot be uniformly applied to all terrestrial invertebrates due to their group-specific and especially species-specific relations to soil microclimatic factors.

We observed significant differences in soil mean temperatures between the doline sites. The sites of circular doline (2), with steep slopes, in particular exhibited larger differences in microclimatic conditions, thus determining larger differences in community parameters of Collembola compared to the other two dolines. Moreover, a gradient of vegetation was visible in this doline, with its bottom occupied by an association typical for wet grasslands in montane and sub-montane regions [65]. Species richness in this doline had a clear pattern, with roughly double the values at the cold doline sites (B, PA) than at the warm site (PB), supported by the negative correlation between this parameter and the maximum values of the soil temperatures. Cold sites, associated with low soil carbon content, showed high abundances (sites PA) and diversity indices (sites PA, B), supported by a negative correlation between the indices and soil parameters.

A significant difference in species richness was also found in the circular doline (3) between the plateau (PA), which was associated with the high content of soil carbon and showed the lowest number of species, and the warm S-facing slope with low carbon content and showing the highest species number, with a considerable proportion of xerothermophilous species. Moreover, significant differences in mean abundance were found between the plateau site (PA) and the cold bottom and warm S-facing slope.

In the relatively large and elliptical doline (1), a reverse community pattern was observed, with increasing species richness and diversity indices towards the surrounding plateau (sites PA and PB), which is in contrast to the previous two dolines, both circular in shape, although no significant differences in community parameters were found between the sites of this doline.

Thus, we may conclude that dolines (2) and (3), both neighbouring and similar in size and morphology, contributed to higher Collembola diversity in this karst area, with higher diversity in these dolines compared to the surrounding plateau. Doline sites, such as cold bottoms and warm S-facing slopes, which were associated with high species richness and abundance of soil Collembola communities, were characterised by low soil carbon content. Soil organic content had a less evident effect on community parameters than soil microclimate. The authors of [18] pointed out the soil microclimate as a key driver of soil Collembola distribution at sites along a microclimatic gradient in a deep karst collapse doline, and [82] found out that both soil microclimate and organic carbon content were drivers of Collembola communities at sites along a microclimatically inversed scree slope in a deep karst valley. In the present study, both environmentally similar dolines also had a very similar community structure, with warm sites (S, PB) associated with xerothermophilous/xeroresistant and warm-adapted species (*H. assimilis, P. bipunctatus* and *S. ununguiculata*). On the other hand, cold sites (B, N, PA) were clearly associated with eurythermic and mesophilous species (*L. cyaneus, M. minimus, P. notabilis, P. pannonica* and *P. albida*).

Eleven species of Collembola that occurred only in the dolines (mostly on the warm S-facing slopes and in the cold bottoms of karst dolines (2) and (3)) occurred in low abundances. Such small populations may be highly endangered by extinction due to climate change. Their exclusiveness also suggests that the studied dolines may facilitate the persistence of some species that are not present or are very rare in the surrounding landscape but support local biodiversity.

*4.3. Distributional Patterns of the Functional Groups*

According to [76], functional trait approaches in ecology are suitable for better understanding the response of soil arthropod communities to environmental gradients. Even when the responses of species composition and diversity in their study were weak, the authors observed that the functional trait composition of the local Collembola community was strongly determined by the topographic gradient. Furthermore, karst dolines have the potential to maintain distinctive arthropod functional groups that are rare or absent in the surrounding habitats (e.g., [8,17,25,28]). Although significant preferences for certain sites were observed only for species numbers, similar patterns for Collembola functional groups were suggested. Hygrophilous and mesophilous groups preferred doline bottoms, and the hygrophilous group also preferred sites on the plateau (PA). As expected, xerophilous/xeroresistant and thermophilous groups clearly preferred the warm sites of the dolines, i.e., the S-facing slopes and sites on the plateau (PA). This study thus showed that dolines can sustain distinctive Collembola functional groups, which reflect microclimatic gradients in the soil within the karst dolines.

*4.4. Endemic and Relict Species vs. Karst Dolines as Microrefugia*

Identification of karst dolines as microrefugia and the presence of endemic and relict species, characterised by narrow distribution ranges, seriously outline the importance of karst dolines from a conservation point of view with the necessity of a sensitive approach to such unique species and their karst habitats. In this study, rare Carpathian and Western Carpathian endemic species were not documented inside of dolines, but they occurred at the plateau sites (PA), slightly N/NE-facing, namely the psychro- and hygrophilous *O. rectopapillatus* and the thermophilous and mesophilous *P. loksai*. Both species are documented from other karst dolines of the Slovak Karst with a more pronounced microclimate inversion [17,20,28]. Generally, a low number of psychrophilous species of Collembola recorded in the present study may interfere with specific topographical and morphological properties of studied dolines, namely their location at lower elevations and the less convex shape and thus gentle slopes, which determine relatively high soil mean temperatures at sites across these dolines (e.g., [8,22,26,27]). The type of vegetation indeed has a significant effect on the distribution of cold-adapted species in dolines. In contrast to open dolines, forest cover captures sunlight better than low-growing grasses and markedly buffers the daily and seasonal temperature variations [83,84]. Therefore, forested dolines are characterised by habitats with lower soil temperatures and marked soil microclimatic stability compared to open habitats (e.g., [27]). The microclimatic stability of cold environments in various karst landforms has been shown by previous studies [11,33,34,85–89]. A higher proportion of cold-adapted Collembola species were recorded in collapse dolines associated with colder and wet sites and with more stable soil temperature variations near the permanent or seasonal ice fillings [18,28,90]. This implies that the type of karst doline associated with the specific geomorphological processes involved [91] can lead to the formation of more or less favourable microclimatic conditions for psychrophilous species.

Currently, while global warming is putting many species in danger of extinction by changing regional temperature and precipitation patterns, topographic variations of the terrain can alter/buffer such effects on a regional scale by creating specific meso-/microclimates, which can provide a complex mosaic of suitable habitats for many species to survive periods of unfavourable climate conditions. Due to the presence of warm and dry habitats in open dolines, these topographically heterogeneous landforms could serve as microrefugia for various xero- or thermophilous species, which could expand as a result of warming in recent decades [92]. On the other hand, the dolines could potentially provide a microclimatic refugium for cold-adapted species through the accumulation of colder air at the bottoms of these landforms, resulting in a buffering of the local microclimate against the surrounding mesoclimate. Similarly, N-facing slopes may thus serve in this way due to the considerably lower insolation and soil temperatures, with assumed lower temperature fluctuations (higher stability) compared to other doline sites [74].

**5. Conclusions**

We found that diverse soil microclimatic conditions within dolines supported higher Collembola diversity (species numbers, diversity indices) compared with sites on the surrounding plateau. In the dolines with circular morphology and of comparable size, topography and soil microclimate had a stronger effect on soil Collembola communities than soil nutrients, such as organic carbon content. Both neighbouring dolines also showed a high degree of similarity in the structure of their soil Collembola communities compared with the larger doline, which is elliptical in shape. The dolines also showed potential to facilitate the persistence of some species that are absent or very rare in the surrounding landscape.

These relatively shallow solution karst dolines provided microhabitats for various functional groups of soil Collembola, with hygrophilous and mesophilous species preferring their cold bottoms and sites on the plateau, while xerophilous/xeroresistant and thermophilous species preferred the warm S-facing slopes and sites on the plateau. The

structure of the communities thus clearly reflected the microclimatic character of the individual sites across the dolines in a north-south direction.

Finally, the karst dolines that provide microclimatically diverse habitats are vulnerable environments in terms of global warming, which may lead to the loss of their refugial potential and consequently to the reduction of biodiversity in these unique natural sites.

**Author Contributions:** Conceptualization, Ľ.K., N.R. and M.M.; methodology Ľ.K., N.R. and M.M.; software, D.M. and J.Š.; validation, M.M., N.R., D.M., J.Š., J.K. and Ľ.K.; formal analysis, M.M., D.M. and J.Š.; investigation, M.M., N.R. and Ľ.K.; data curation, M.M. and D.M.; writing—original draft preparation, M.M.; writing—review and editing, N.R., D.M., J.Š., J.K. and Ľ.K.; visualization, M.M., D.M. and J.Š.; supervision, N.R. and Ľ.K.; project administration, Ľ.K.; funding acquisition, Ľ.K. All authors have read and agreed to the published version of the manuscript.

**Funding:** This research was funded by the Slovak Scientific Grant Agency, grant number VEGA 1/0346/18 and 1/0438/22 and the Slovak Research and Development Agency, grant number APVV-21-0379.

**Institutional Review Board Statement:** Not applicable.

**Data Availability Statement:** The datasets generated during the current study are available on reasonable request.

**Acknowledgments:** P. Ľuptáčik (P. J. Šafárik University in Košice, Košice, Slovakia) is acknowledged for assistance during the field work. R. Šuvada (Administration of the Slovak Karst National Park, Brzotín, Slovakia) is acknowledged for the analysis of the vegetation associations of the studied dolines. We are grateful to D. L. McLean for linguistic correction of the manuscript and two reviewers for their constructive comments.

**Conflicts of Interest:** The authors declare no conflict of interest. The funders had no role in the design of the study; in the collection, analyses, or interpretation of data; in the writing of the manuscript; or in the decision to publish the results.

## Appendix A

**Table 1.** Mean abundance (ind.m$^{-2}$) of Collembola species at sites in three open dolines.

| Ecol. Category | Species | 1PA | 1N | 1B | 1S | 1PB | 2PA | 2N | 2B | 2S | 2PB | 3PA | 3N | 3B | 3S | 3PB |
|---|---|---|---|---|---|---|---|---|---|---|---|---|---|---|---|---|
| e[1,2], m[3] | *Allacma fusca* (Linné, 1758) | 25 | – | 25 | – | 76 | – | 25 | – | 76 | – | 51 | – | – | – | – |
| t[4,5], x[4,6] | *Axenyllodes bayeri* (Kseneman, 1935) | – | – | – | – | – | – | – | – | – | – | 51 | – | – | – | – |
| e[4,7], m[7] | *Ceratophysella engadinensis* (Gisin, 1949) | – | – | – | – | – | 331 | – | 153 | – | – | – | – | – | – | – |
| e[7], m[7] | *Ceratophysella succinea* (Gisin, 1949) | – | – | – | – | 204 | 1605 | – | 510 | – | – | 25 | – | 25 | – | 204 |
| e[8], m[8] | *Desoria germanica* (Hüther et Winter, 1961) | 382 | 204 | 586 | 25 | 611 | – | 280 | 306 | 25 | – | – | 204 | 25 | 255 | 127 |
| e[9], m[9] | *Deutonura conjuncta* (Stach, 1926) | 25 | – | – | – | – | – | – | – | – | – | – | – | 25 | – | – |
| t[10,11], x[10,11] | *Doutnacia mols* Fjellberg, 1998 | – | – | – | – | – | 408 | – | – | – | – | – | – | – | – | – |
| t[4,10], x[4,10] | *Doutnacia xerophila* Rusek, 1974 | 204 | 25 | 25 | 102 | 25 | 2981 | 51 | 25 | 25 | – | – | 153 | 25 | 25 | 866 |
| t[12], x[12,13] | *Entomobrya handschini* Stach, 1922 | – | – | – | – | – | – | – | – | – | – | 25 | – | – | – | – |
| t[12], m[13] | *Entomobrya quinquelineata* Börner, 1901 | – | – | – | – | – | – | 25 | – | 25 | – | – | – | – | – | – |
| un | *Entomobrya* sp. 1 | – | – | – | – | 51 | 25 | 76 | 76 | 51 | 76 | 178 | 357 | 25 | 76 | 127 |
| un | *Entomobrya* sp. 2 | 331 | 76 | 51 | 102 | – | 25 | 51 | – | – | – | 25 | 51 | – | – | – |
| e[4,8], m[8] | *Folsomia manolachei* Bagnall, 1939 | – | – | 892 | 459 | 76 | – | – | – | – | – | – | – | 51 | 76 | – |
| e[4,8], m[13] | *Folsomia penicula* Bagnall, 1939 | – | – | – | 25 | – | – | – | – | – | – | – | – | – | – | – |
| e[13,14], h[13] | *Folsomia quadrioculata* (Tullberg, 1871) | 331 | 102 | 229 | – | – | – | – | – | – | – | – | – | – | – | – |
| e[14], m[13] | *Friesea mirabilis* (Tullberg, 1871) | – | – | – | – | – | – | – | – | – | – | – | 25 | – | – | – |
| e[14,15], m[13] | *Friesea truncata* Cassagnau, 1958 | – | – | 255 | – | – | – | – | 1147 | – | – | – | – | – | – | – |
| t[8,16], x[13] | *Hemisotoma thermophila* (Axelson, 1900) | – | – | – | – | – | 1656 | – | – | – | – | – | – | – | – | – |
| t[17], m[13] | *Heteromurus nitidus* (Templeton, 1835) | 127 | 25 | – | 127 | 25 | 51 | 331 | 331 | 153 | – | – | 76 | 76 | 153 | 25 |
| e[7,18], m[19] | *Hypogastrura assimilis* (Krasusbauer, 1898) | 459 | 3440 | 76 | – | 1325 | 1197 | 5885 | 1045 | 4433 | 9376 | 1096 | 1707 | 51 | 13554 | 10650 |
| t[4,8], x[8,20] | *Isotoma anglicana* Lubbock, 1862 | 204 | 204 | 357 | 586 | 535 | 1197 | 255 | 153 | – | 484 | 484 | 357 | 280 | 76 | 459 |
| e[21], m[8,22] | *Isotomiella minor* (Schäffer, 1896) | 1605 | 51 | 841 | 51 | – | – | 1299 | 1758 | – | – | – | 2064 | 15593 | 2930 | 102 |
| t[4,8], x[4,8] | *Isotomodes productus* (Axelson, 1906) | – | – | – | – | 306 | 102 | – | – | – | – | 25 | – | – | – | 76 |
| e[10], m[10] | *Karlstejnia annae* Rusek, 1974 | 153 | – | – | – | 51 | – | – | – | – | – | – | – | – | – | – |
| e[13], m[13] | *Lepidocyrtus cyaneus* Tullberg, 1871 | 229 | 51 | 866 | 943 | 561 | 943 | 5580 | 1860 | 255 | 433 | – | 917 | 1427 | 2497 | 102 |
| e[20], m[20] | *Lepidocyrtus lignorum* (Fabricius, 1775) | – | – | – | – | – | 51 | – | 25 | – | – | – | 25 | 611 | 76 | – |
| e[23], m[23] | *Megalothorax minimus* Willem, 1900 | 1809 | 306 | – | 1045 | 102 | 535 | 1503 | 459 | 153 | 204 | 25 | 306 | 4790 | 382 | 51 |
| e[24], m[24] | *Megalothorax willemi* Schneider et d'Haese, 2013 | 51 | – | – | 102 | – | 51 | – | – | – | – | – | – | 25 | – | – |
| t[4,10], m[13] | *Mesaphorura critica* Ellis, 1976 | 153 | 204 | 178 | 102 | 127 | 2369 | 76 | 102 | 102 | 25 | 229 | 25 | 357 | 51 | 1248 |
| e[2,4], m[10] | *Mesaphorura florae* Simón, Ruiz, Martin et Luciáněz, 1994 | 76 | – | – | 127 | – | – | 51 | – | – | – | – | – | – | 102 | 178 |
| e[10], m[13] | *Mesaphorura hylophila* Rusek, 1982 | – | – | 25 | – | – | 76 | – | 25 | – | – | – | – | – | – | – |
| e[4], m[14] | *Mesaphorura italica* (Rusek, 1971) | – | – | – | – | – | 25 | 25 | 25 | – | – | – | 51 | – | – | – |
| e[10], m[10] | *Mesaphorura jirii* Rusek, 1982 | – | – | – | – | – | – | – | – | – | – | – | – | 76 | – | 51 |
| un | *Mesaphorura rudolfi* Rusek, 1987 | – | – | – | – | – | – | – | – | – | – | – | – | – | 51 | – |
| e[10], m[10] | *Mesaphorura yosii* (Rusek, 1967) | – | 25 | – | – | 25 | – | – | – | – | 25 | – | – | 51 | 25 | – |
| t[10,25], x[4,10] | *Metaphorura affinis* (Börner, 1902) | – | – | 25 | – | 76 | 306 | – | – | 127 | – | – | – | – | – | – |
| e[26], m[23,26] | *Micranurida pygmaea* Börner, 1901 | – | – | – | – | – | 51 | – | – | – | – | – | – | – | – | – |
| t[27], x[27] | *Microgastrura duodecimoculata* Stach, 1922 | 739 | – | – | – | – | – | – | – | 102 | – | – | – | – | 51 | – |

**Table 1.** *Cont.*

| Ecol. Category | Species | 1PA | 1N | 1B | 1S | 1PB | 2PA | 2N | 2B | 2S | 2PB | 3PA | 3N | 3B | 3S | 3PB |
|---|---|---|---|---|---|---|---|---|---|---|---|---|---|---|---|---|
| e [2,4], m [13] | *Oncopodura crassicornis* Shoebotham, 1911 | – | – | – | 25 | 51 | – | – | – | – | – | – | – | 25 | 51 | – |
| e [4], m [28] | *Orchesella bifasciata* Nicolet, 1842 | – | – | – | – | – | – | – | 76 | – | – | – | – | – | – | – |
| t [29], x [13,29] | *Orchesella multifasciata* Stscherbakow, 1898 | – | – | – | 25 | – | – | 25 | 102 | 51 | – | – | 25 | 178 | – | – |
| c [4,30], h [4,30] | *Orthonychiurus rectopapillatus* (Stach, 1933) | – | – | – | – | – | – | – | – | – | – | 25 | – | – | – | – |
| e [8], m [8] | *Parisotoma notabilis* (Schäffer, 1896) | 790 | 535 | 739 | 408 | 25 | 1096 | 229 | 815 | 25 | 51 | 306 | 1325 | 1045 | 25 | – |
| t [14], x [13,20] | *Pratanurida cassagnaui* Rusek, 1973 | 76 | 408 | – | 25 | – | 76 | – | 51 | – | 51 | 204 | 459 | 102 | 102 | 76 |
| e [8], m [8] | *Proisotoma brevidens* Stach, 1947 | – | 76 | – | – | – | – | – | – | – | – | – | – | – | – | – |
| t [8,31], x [8,31] | *Proisotomodes bipunctatus* (Axelson, 1903) | – | – | – | – | – | – | – | – | 2573 | 994 | – | – | – | – | – |
| e [4,30], m [4,30] | *Protaphorura armata* (Tullberg, 1869) | – | 51 | 127 | 2420 | 25 | 25 | 331 | – | – | – | – | – | 229 | – | – |
| e [30], m [30] | *Protaphorura pannonica* (Haybach, 1960) | 943 | 968 | 229 | 204 | 1045 | 4255 | 1121 | – | 306 | 127 | – | 2217 | 4561 | 510 | 1554 |
| t [4], m [13] | *Protaphorura serbica* (Lokša et Bogojevič, 1967) | – | – | – | – | – | – | – | – | 153 | – | – | – | – | – | – |
| t [4,5], x [4,13] | *Pseudachorutes pratensis* Rusek, 1973 | 25 | – | – | – | 713 | 76 | 51 | 25 | 76 | – | – | – | – | 51 | 51 |
| e [2], m [2] | *Pseudosinella albida* (Stach, 1930) | 1783 | 943 | 2854 | 484 | 102 | 127 | 280 | 1197 | 739 | – | 76 | 153 | 25 | 25 | – |
| un | *Pseudosinella* cf. *csafordi* Winkler et Mateos, 2018 | – | – | – | – | – | – | – | – | – | – | – | – | 76 | 25 | – |
| e [2], m [13] | *Pseudosinella horaki* Rusek, 1985 | 892 | – | 102 | 1401 | 25 | – | 178 | 892 | 25 | – | 25 | – | 51 | 25 | – |
| t [32], m [2] | *Pumilinura loksai* (Dunger, 1973) | 102 | – | – | – | – | – | – | – | – | – | – | – | – | – | – |
| e [7], x [25] | *Schoettella ununguiculata* (Tullberg, 1869) | – | – | – | – | – | – | – | – | – | 102 | – | – | – | 4408 | – |
| e [1], m [20] | *Sminthurinus aureus* (Lubbock, 1862) | 179 | – | 102 | – | – | 204 | 25 | 127 | 561 | 153 | 51 | 331 | – | – | – |
| t [2], m [1,13] | *Sminthurinus elegans* (Fitch, 1863) | 25 | 178 | 841 | 331 | 586 | – | 382 | 357 | – | – | 76 | – | – | 204 | 102 |
| t [1], x [1] | *Sminthurus maculatus* Tömösváry, 1883 | – | – | – | – | 51 | – | – | – | – | – | – | – | – | – | – |
| e [1], m [13] | *Sphaeridia pumilis* (Krausbauer, 1898) | 102 | – | – | – | – | – | – | 25 | 459 | – | 25 | 51 | 1096 | – | – |
| un | *Wankeliella* sp. juv. | – | – | – | – | 25 | – | – | – | – | – | – | – | – | 25 | 25 |
| t [14], x [14] | *Willemia scandinavica* Stach, 1949 | – | 51 | – | – | – | 306 | 127 | – | – | – | 25 | 153 | – | – | 280 |
| t [20], x [6,20] | *Willowsia buski* (Lubbock, 1869) | – | – | – | – | – | – | – | 25 | 51 | – | – | – | – | 178 | – |
| t [20], x [13,20] | *Willowsia nigromaculata* (Lubbock, 1873) | – | – | – | – | – | – | – | – | – | – | – | – | – | 382 | – |

Ecological category: c—cold-adapted species, t—thermophilous, e—eurythermic, h—hygrophilous, m—mesophilous, x—xerophilous/xeroresistant, un—unclear (For site description, see the Section 2); 1—[54]; 2—[20]; 3—[93]; 4—[17]; 5—[60]; 6—[56]; 7—[58]; 8—[57]; 9—[94]; 10—[95]; 11—[96]; 12—[61]; 13—[97]; 14—[55]; 15—[98]; 16—[99]; 17—[100]; 18—[101]; 19—[102]; 20—[103]; 21—[104]; 22—[105]; 23—[106]; 24—[107]; 25—[108]; 26—[109]; 27—[110]; 28—[111]; 29—[112]; 30—[113]; 31—[114]; 32—[115].

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
