# Peer review of "Karst Dolines Support Highly Diversified Soil Collembola Communities—Possible Refugia in a Warming Climate?"

_diversity, doi:10.3390/d14121037_

Round 1
Reviewer 1 Report
I started to read this paper with high expectations as the aim as stated had considerable interest and conservation significance i. e. that of whether dolines acted as conservation reservoirs for species that were apparently absent or rare on the surrounding plain. Unfortunately the contents did not meet my hopes. The English and grammar were adequate but the method was faulty. I attempt to explain my criticisms below.
1. I particularly object to the term "functional species". Functional biodiversity is worse. What on earth does it mean.? please do not explain, just remove it! This is because we cannot know what the function of a species is. I know it is a popular term and sadly much used but it is a misnomer. You can use the term ecological which is better but still not exactly true. Please remove it from the text.
2. My main complaint is a criticism of the method. The authors assume the base of the doline is more humid and cool than the plain above, that is fine but they cannot then classify the species into humid loving or not before they do the analysis. This is tantemount to saying I am going to test whether species A which I know is humid loving is commoner in the damp bottom of the doline and surprise surprise I find it is. This is not an objective test of the hypothesis. The authors must not make assumptions about species before analysis as they are applying a bias to the analysis. So, the authors must re-analise their data without deciding before hand on the ecological preferences of any species.
2. Another problem I noticed is that some results are reported in the text while they would be better understood if displayed in histograms or similar type of graphs.
3. A more minor point is that the authors have used words already in the title for key words. This is not allowed as it is repetition. There are four words of this type.
4. There is no such word as Collembolans. Yuk! Collembola is a perfectly good plural, collembolan is an adjective with lower case c, The singular is Collembolon.
5. In my printed version the maps in figure 1 are too small.
6. Taking five soil cores is not taking five replicate samples. That is pseudoreplication.
7. High gradient extraction of soil cores is one of the most inefficiant methods of collecting animals from soil. Read the literature more critically!
8, Why use a proxy for soil moisture? Why not measure it?
9. Cryptopygus thermophilus is an old incorrect name.
10. On first mentioning a species name in the text you must add the authority,
11. Figure 4 was too crowded so impossible to interpret. Put sites and species on separate graphs.
12. Paragraph beginning line 335 is an example of the bias I described in the method i.e. authors say here they put in A, B and C and surprise surprise they got out A, B and C. Analysis must be redone objectively.
13. Discussion does not exactly stress that the results are new and make a significant contribution to survival of rare species in dolines. Maybe they do but I did not get this message from this discussion. I think it would have been more productive to classify each species according to their distribution i.e. cosmopolitan, regionally restricted, endemic and locally endemic and compare patterns.
14. The last sentence mentions that dolines harbour species with narrow distribution ranges but I did not see this data presented in the results clearly.
I did not check the refences.
Reviewer 2 Report
Line 2-3 Title Karst dolines support highly diversified soil Collembola communities – possible refugia in a warming climate?
I suggest “Do Karst dolines support highly diversified soil Collembola communities – possible refugia in a warming climate?”
Line 173-174 Ranunculo bulbosi-Arrhenatheretum elatioris association, perhaps Author Ranunculo bulbosi-Arrhenatheretum elatioris Ellmauer in Mucina et al. 1993
Line 175 Brachypodio pinnati-Molinietum arundinaceae Klika 1939
Lines 176-177 Festuco rupicolae-Nardetum strictae Dostál 1933
Lines 178 Onobrychido viciifoliae-Brometum erecti (Scherrer 1925) Müller 1966
Line 179 association, must be not in italics: association,
Lines 179-180 Scabioso ochroleucae-Brachypodietum pinnate must be Scabioso ochroleucae-Brachypodietum pinnati Klika 1933
Line 1980 Rosetum gallicae Kaiser 1926
Line 182 Alchemillo-Arrhenatheretum elatioris Sougnez and Limbourg 1963
Line 183 Festuco rupicolae-Caricetum humilis Klika 1939
Line 184 Onobrychido viciifoliae-Brometum erecti (Scherrer 1925) Müller 1966
Line 185 Scabioso ochroleucae-Brachypodietum pinnati Klika 1933
Line 186 Pastinaco sativae-Arrhenatheretum elatioris Passarge 1964
Table 2 Doline 3 is the smallest in diameter, and the shallowest, however, it is the one with the most Collembola abundance at the bottom.
Could you venture any explanation?
Lines 459-460 “Hygrophilous and mesophilous groups preferred doline bottoms,
while the hygrophilous group also preferred sites on the plateau (PA)”.
There seems to be some contradiction in this sentence.
Round 2
Reviewer 1 Report
I still reject this mss in its current form. The authors have not accepted my recommendations and state I am wrong! In particular they insist in using the term function etc without providing an adequate reason. I stated that we cannot know what a function of a species is so to use this term is just incorrect. Also their method is inefficient. High gradients fry some species before they can escape. Certainly if intact cores are extracted. This is well known. Just compare crumbling soil cores and spreading them thinly in a funnel. AND they do not seem to have understood the bias they have used in their method So I continue to reject this mss.
